# Box breathing or six breaths per minute: Which strategy improves athletes post-HIIT cardiovascular recovery?

**Murat Kasap** [ID]*, **Gökhan Recep Aydin**

Department of Coach Education, Bartın University, Faculty of Sports Sciences, Bartın, Türkiye

* mkasap@bartin.edu.tr

## Abstract

Post-exercise recovery strategies are critical for athletic performance, yet the acute effects of controlled breathing techniques (box breathing vs. 6 bpm [6 breaths/min]) following high-intensity interval training (HIIT) remain understudied. This study compared three breathing protocols' impact on cardiovascular and perceptual recovery metrics. In a randomized crossover design, 40 physically active university students (25 males, 15 females; age 20.95 ± 1.75 years) completed three HIIT sessions on a spin bike (15 min, 1:2 work: rest ratio at 85–95% HRmax). One of three recovery breathing protocols was applied during each session: Spontaneous breathing (control), Box breathing (4-4-4-4), 6 bpm (5–5). Heart rate (HR) was monitored continuously, and perceived exertion was assessed via Borg Scale (6–20). Data were analyzed using repeated-measures ANOVA and Tukey's HSD post-hoc tests (α = 0.05). Box breathing elicited significantly: Higher post-exercise HR (164.65 ± 9.40 bpm) vs. 6 bpm (154.77 ± 12.18 bpm; $p < 0.001$*, Cohen's d = 0.907*) and spontaneous breathing (159.05 ± 10.38 bpm; $p = 0.054$*), Elevated perceived exertion (Borg 17.27 ± 1.30) vs. 6 bpm (15.25 ± 1.08; $p < 0.001$*) and spontaneous breathing (15.25 ± 1.35; $p < 0.001$*). No significant difference in recovery time was observed ($p = 0.128$*), though box breathing showed a trend toward delayed HR baseline recovery (753.02 ± 150.60 sec vs. 675.70 ± 157.24 sec with 6 bpm). While 6 bpm appears optimal for post-HIIT recovery, box breathing may impose physiological and perceptual stress. Practitioners should tailor breathing strategies to individual tolerance and exercise intensity. These findings highlight the need for personalized recovery protocols in athletic training.

## 1. Introduction

The post-exercise recovery process plays a critical role in increasing training efficiency, maintaining high-level performance, and reducing the risk of injury in athletes [1,2]. Modern training science considers recovery to be one of the fundamental determinants

**Data availability statement:** All relevant data are within the manuscript and its Supporting Information files.

**Funding:** The author(s) received no specific funding for this work.

**Competing interests:** The authors have declared that no competing interests exist.

of performance, taking into account not only the training load but also the manner and speed with which the body responds to it. Recovery is seen as a window through which the body's physiological and neurovegetative (autonomic nervous system) adaptations to acute exercise stress can be assessed [3]. In this context, the recovery rate of the cardiovascular system specifically, the time required for the heart rate (HR) to return to baseline levels after exercise is considered a critical biomarker for assessing training intensity and individual physiological capacity [4]. In addition to traditional recovery strategies such as nutrition, cooling, active rest, and hydrotherapy, controlled breathing protocols have gained increasing scientific interest in recent years [5,6].

Studies on the physiological effects of breathing exercises have reported positive results in cardiopulmonary parameters such as increased vagal tone, improved heart rate variability (HRV), and improved baroreflex sensitivity [7,8]. Slow and controlled breathing suppresses sympathetic activity and increases parasympathetic responses, leading to significant improvements in heart rate, and blood pressure. This physiological shift is directly associated with a reduction in psychological stress, which in this context refers to measurable decreases in perceived stress and state anxiety, as well as an improvement in overall emotional well-being [9,10,11].

Box Breathing and 6 bpm protocols are prominent applications in this field. Box Breathing consists of a four-stage cycle, each lasting four seconds (inhale→hold→exhale→hold), and is commonly used by military personnel and athletes to enhance focus under stress and maintain physiological balance [3,12,13]. The 6 bpm protocol, also known as resonance frequency breathing, has been shown to optimize cardiorespiratory synchronization [14,11]. The positive effects of these techniques on the heart and autonomic systems have been confirmed in numerous studies. For example, Kumar et al. [2] reported that six breaths per minute (6 bpm) significantly reduced anxiety and heart rate. Bentley et al. [15] identified moderate effect sizes in psychological variables. Zaccaro et al. [1] observed significant reductions in cortisol levels and heart rate following slow breathing interventions. Joseph et al. [5] noted that controlled breathing increases baroreflex sensitivity and lowers blood pressure in hypertensive individuals.

However, much of the existing literature on breathing techniques has focused on clinical populations or psychological stress outcomes [12,15]. In particular, studies directly examining the effects of such breathing protocols on heart rate recovery time and perceived exertion following high-intensity interval training (HIIT) are quite limited.

Building on this gap, this study aimed to investigate the acute effects of three different breathing protocols (Box Breathing, 6 bpm, Spontaneous Breathing) on physiological recovery during high-intensity interval (HIIT) cycling exercise on a spin bike in sports science students of similar age and physical capacity.

The hypotheses tested in the study were as follows: *H1:* Box Breathing and 6 bpm protocols will positively affect heart rate recovery and perceived exertion (Borg RPE scores) compared to spontaneous breathing. *H2:* Controlled breathing techniques will reduce perceived exertion (Borg RPE scores) more effectively than spontaneous breathing. *H3:* 6 bpm may have a stronger effect on cardiovascular and perceptual recovery metrics than the Box Breathing protocol.

## 2. Materials and methods

### Participants

Forty students from Bartın University's Faculty of Sports Sciences (25 male, 15 female; age: 20.95±1.75 years; BMI: 22.97±3.25 kg/m²) voluntarily participated in the study. The participants were athletes who engaged in physical activity comprising at least two days of aerobic or anaerobic exercise per week, had no prior experience with breathing exercises or meditation-based activities, and were not currently practicing them. All participants were familiar with cycle ergometer training and had no known cardiopulmonary or orthopedic disorders. The participants were not practicing any techniques such as single-breath diving, breathing exercises, or meditation. The study was approved by the Bartın University Ethics Committee (Approval No: 2025-SBB-0244) and conducted in accordance with the Declaration of Helsinki. All participants were of legal age and provided written informed consent prior to the study.

### Inclusion Criteria:

- Voluntary participation.

- Age 18–24.

- Student in the Faculty of Sports Sciences.

- Must not have previously engaged in, and must not be currently engaging in, any breathing exercises or meditation-based activities.

### Exclusion Criteria:

- Failure to meet one or more of the inclusion criteria listed above was defined as an exclusion criterion.

### Randomization

The order of the three breathing protocols was randomized for each participant using a computer-generated random sequence (www.randomizer.org).

*The order was as follows:*

Session 1

37, 36, 11, 27, 15, 38, 1, 29, 24, 7, 4, 2, 34, 23, 33, 6, 14, 5, 40, 21, 12, 32, 13, 3, 16, 25, 31, 35, 18, 26, 30, 17, 20, 9, 8, 28, 19, 39, 10, 22

Session 2

32, 2, 19, 37, 33, 11, 15, 21, 29, 39, 28, 6, 40, 31, 18, 27, 22, 16, 3, 38, 12, 13, 26, 4, 30, 35, 36, 17, 34, 5, 20, 10, 24, 7, 25, 14, 23, 8, 9, 1

Session 3

5, 12, 15, 10, 28, 13, 4, 36, 25, 31, 18, 19, 2, 7, 35, 39, 32, 3, 40, 14, 26, 23, 22, 33, 9, 38, 16, 29, 8, 17, 6, 34, 11, 20, 1, 37, 24, 30, 21, 27.

### Measured and observed parameters

The following primary outcome variables were assessed for each breathing protocol:

- Initial Heart Rate (HR): HR measured at the end of the 10-minute warm-up period.

- Post-Exercise Heart Rate (HR): HR measured immediately upon completion of the final HIIT interval.

- Heart Rate Recovery Time: The time elapsed from the cessation of exercise until HR returned to the individual's pre-exercise baseline (Initial HR) value.

- Rating of Perceived Exertion (RPE): Assessed using the Borg RPE scale (6–20) immediately after each test session.

## Determination of HRmax

Maximum heart rate (HRmax) was estimated using the Tanaka formula: HRmax = 208 − (0.7 × age), which has been shown to be more accurate than the traditional 220 − age formula, particularly in healthy adults [16]. This estimation guided the target training zone (85–95% HRmax) during HIIT intervals.

## Control of respiratory protocols

Specially designed video slides for the three distinct breathing protocols were projected onto the wall in front of the participants in the spin bike room. The test flow guided the athletes both visually and auditorily (Fig 1).

Respiratory control was rigorously ensured through a standardized, audiovisual guidance system that minimized participant discretion and ensured strict adherence to the prescribed breathing protocols.

*The procedure was as follows:*

Specially designed video slides for each of the three distinct breathing protocols were projected onto the wall in front of the participants. The entire test flow was governed by this system, which provided simultaneous visual and auditory commands.

Prior to each testing session, participants were familiarized with the specific breathing protocol they would perform to ensure comprehension.

*The control process for a single cycle (e.g., the 4-4-4-4 box breathing protocol) was as follows:*

1- The session began with a 30-second exercise bout initiated by an audible "start test" command. A countdown timer was displayed. At the end of the exercise, an auditory cue announced the transition to a 60-second rest/breathing period. During this period, each phase of the breathing cycle was precisely guided:

2-3-4-5 An audible "inhale" cue was given, accompanied by a visual timer counting down 4 seconds. This was sequentially followed by "hold breath" (4 seconds), "exhale" (4 seconds), and again "hold breath" (4 seconds). This cycle was repeated throughout the 60-second rest period. A verbal and visual warning was given at the 55-second mark to prepare for the next exercise bout.

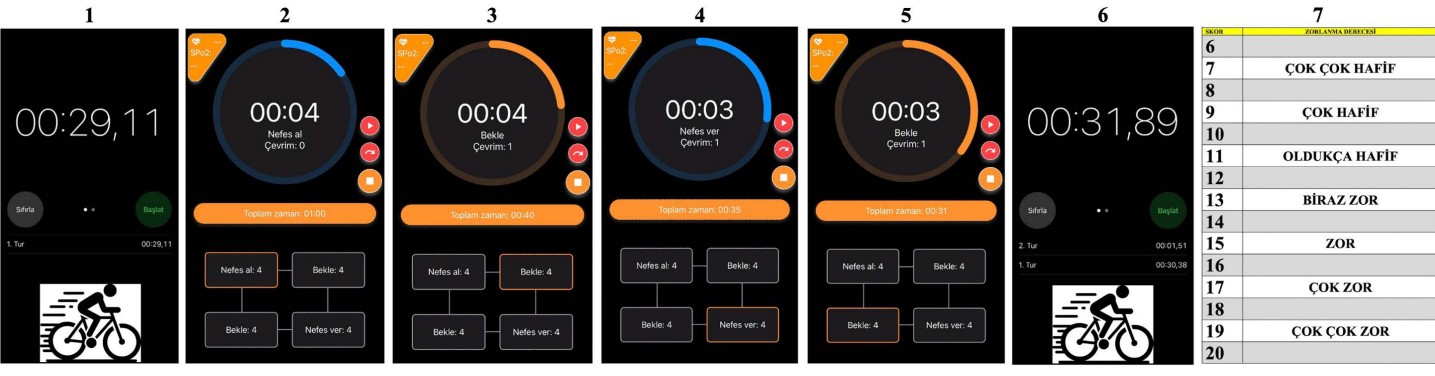

**Fig 1. Example (4-4-4-4) box breathing video presentation stage visuals.**

Moreover "Researchers visually monitored each participant's chest movement and provided immediate verbal feedback to ensure adherence to the target breathing rhythm throughout the recovery period."

The sequence of 30-second exercise followed by the 60-second guided breathing period was repeated until the total session duration of 15 minutes was completed.

6- The test was terminated automatically upon reaching the 15-minute mark with an audible "test finished" command, immediately followed by the display of the Borg Scale for participant rating.

This method of automated, step-by-step audiovisual guidance ensured that all participants followed the exact same rhythm and timing for each breath, thereby guaranteeing adequate and consistent respiratory control throughout the procedures.

### Application procedure

A randomized crossover design was employed in the study. Each participant completed three separate sessions, applying different breathing protocols during recovery. Data collection took place between April 21 and April 30, 2025, after obtaining ethics committee approval. The order of the sessions was randomly determined, and a minimum of 48 hours of rest was allowed between sessions. During the recovery phase, only the breathing protocol was varied, while all other conditions were kept constant. (The protocol design is illustrated in Fig 2).

### Exercise protocols

Warm-up: 10 minutes of steady-state cycling.

Main exercise: 15 minutes of high-intensity interval training (HIIT) 1:2 (30 seconds of work/60 seconds of rest). Exercise intensity: 100W constant resistance at 85–95% of each participant's maximum heart rate (HRmax). (Monitored using a Polar H10 (KEMPELE, Finland) chest strap and mobile app) (Heart rate response during the interval protocol is shown in Fig 3).

### Breathing protocols

*Spontaneous Breathing:* No specific breathing pattern was applied (this served as the control condition).

*Box Breathing (4-4-4-4):* Inhale (4 seconds) → hold (4 seconds) → exhale (4 seconds) → hold (4 seconds).

*6 bpm (6 Breaths per Minute) [5–5]:* Inhale through the nose (5 seconds) → exhale through the mouth (5 seconds).

* Participants underwent a training process before applying each protocol. Prior to application, each participant's ability to apply the relevant breathing protocol correctly and consistently for 2 minutes was verified by direct observation by the researchers. Where necessary, training was repeated and the protocol was internalized before proceeding to the test.

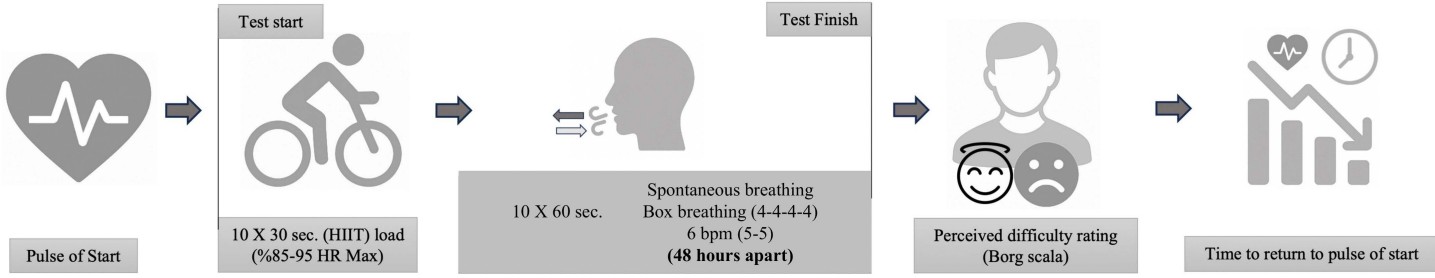

**Fig 2. Flowchart of the Exercise Test Protocol.**

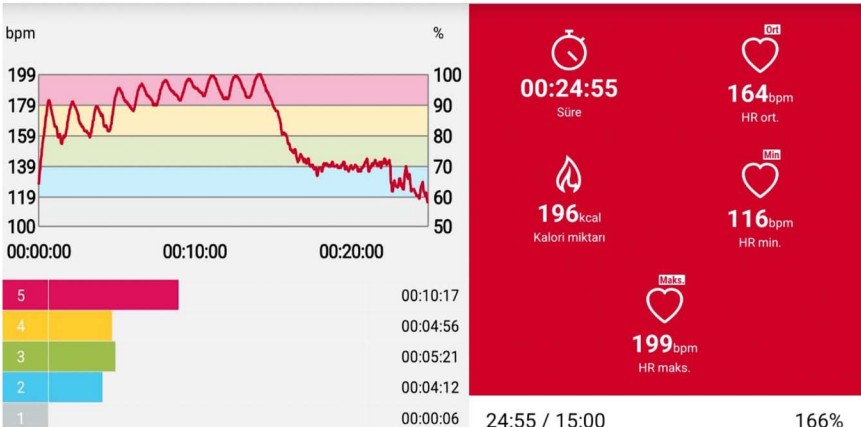

**Fig 3. Heart Rate-Based Intensity Monitoring During Interval Exercise.** Polar H10 (KEMPELE, Finland) mobile app training summary.

## Anthropometric measurements

The height and body weight of participants were measured using a SECA (Germany) scale with an integrated stadiometer. The device has an accuracy of ±0.01 mm for height and ±0.1 kg for weight. Body Mass Index (BMI) values were determined using bioelectrical impedance analysis (BIA) with a TANITA BC420 device. A specific electrical current (50 kHz) was applied through the foot electrodes, and the results were recorded.

## Modified borg scale (rating of perceived exertion)

The Borg RPE (Rating of Perceived Exertion) scale was developed by Gunnar Borg in 1970 to assess the intensity of physical effort during exercise. It is commonly used to evaluate both exertional and resting dyspnea severity [17]. The scale consists of 6–20 graded descriptors corresponding to perceived exertion levels. After each test session, participants were presented with a printed A4 sheet displaying the modified Borg scale and asked to rate their perceived physical effort on the 6–20 scale. The scores were recorded along with the participant's test data.

*Initial Heart Rate:* Heart rate measured at the end of the 10-minute warm-up period before the spin bike exercise.

*Post-Exercise Heart Rate:* Heart rate measured immediately after the final interval was completed.

*Recovery Time:* The time elapsed from the end of the exercise until the heart rate returned to its baseline value.

*Perceived Exertion:* Assessed post-exercise using the Borg RPE scale (6–20).

## Statistical analysis

Descriptive statistics are presented as mean ± standard deviation (SD) along with minimum and maximum values. The normality of the data distribution was assessed using the Shapiro-Wilk test. Since all dependent variables (post-exercise heart rate, time to return to baseline, and perceived exertion score) showed a normal distribution for the three breathing protocols (spontaneous, box breathing, and 6 bpm), parametric tests were used. Differences between protocols were analyzed using the ANOVA test for one-way repeated measures. When significant main effects were found, Tukey's HSD post-hoc test was applied to determine differences between group pairs. All statistical analyses were performed using IBM SPSS Statistics 26.0 software, and the significance level was set at $p < 0.05$.

The sample size of the study was determined using a priori power analysis conducted with the G*Power 3.1 statistical software [18]. According to the analysis, for a three-condition one-way repeated measures ANOVA test, assuming a medium-high effect size (Cohen's d ≈ 0.738; f ≈ 0.38) [19] and a significance level of 0.05, a statistical power of approximately 0.995 (1–β) was obtained with 40 participants (See Table 5. for effect sizes).

## 3. Results

### Participant characteristics

A total of 40 sports science students (25 males, 15 females) participated in the study. The mean age was 20.95 ± 1.75 years, mean height was 173.28 ± 7.44 cm, mean body weight was 69.20 ± 11.83 kg, and mean body mass index (BMI) was 22.97 ± 3.25 kg/m². The minimum and maximum values for each parameter are presented in (Table 1).

### Normality of data distribution

Shapiro-Wilk tests showed that all outcome variables (final heart rate, time to return to baseline, and Borg scale) for each of the three breathing protocols were normally distributed (p > 0.05, Table 2). Therefore, parametric tests were applied in subsequent analyses.

### Main effects of respiratory protocols

Table 3 presents the mean ± SD values for final heart rate, return time to baseline heart rate, and perceived exertion score (Borg Scale) in three respiratory protocols (Spontaneous, Box breathing, and 6 bpm). Repeated measures ANOVA

**Table 1. Descriptive Statistics of Participants' Demographic Characteristics (Mean ± SD).**

|  | Min. | Max. | Mean | SD |
|---|---|---|---|---|
| Age (years) | 18.00 | 24.00 | 20.95 | 1.75 |
| Height (cm) | 160.00 | 189.00 | 173.28 | 7.44 |
| Body Weight (kg) | 46.00 | 94.00 | 69.20 | 11.83 |
| BMI (kg/m²) | 17.10 | 30.48 | 22.97 | 3.25 |

Plus-minus values are presented as mean ± standard deviation (SD). Minimum and maximum values are also given for each variable.

**Table 2. Normality Distribution (Shapiro-Wilk Test Results).**

| Breathing Protocol | Variable | p- value | Normal Distribution |
|---|---|---|---|
| Spontaneous Breathing | Final HR | 0.3100 | Yes |
|  | Return Time to Baseline HR | 0.1973 | Yes |
|  | Borg Scale | 0.3172 | Yes |
| Box Breathing (4-4-4-4) | Final HR | 0.2208 | Yes |
|  | Return Time to Baseline HR | 0.7900 | Yes |
|  | Borg Scale | 0.3845 | Yes |
| 6 bpm Breathing (5−5) | Final HR | 0.2803 | Yes |
|  | Return Time to Baseline HR | 0.4983 | Yes |
|  | Borg Scale | 0.1362 | Yes |

Normality was assessed using the Shapiro-Wilk test. A p-value > 0.05 was interpreted as an indicator of normal distribution. All variables in each respiratory protocol were found to show normal distribution. HR: Hearth Rate.

**Table 3. Descriptive Statistics and ANOVA Results by Breathing Protocol.**

| Variable | Spontaneous Breathing | Box Breathing (4-4-4-4) | 6 bpm Breathing (5−5) | F | p-value |
|---|---|---|---|---|---|
| Final HR (bpm) | 159.05 ± 10.38 | 164.65 ± 9.40 | 154.77 ± 12.18 | 8.54 | **< 0.001** |
| Return Time to Baseline HR (sec) | 703.05 ± 202.71 | 753.02 ± 150.60 | 675.70 ± 157.24 | 2.08 | 0.128 |
| Perceived Exertion (Borg Scale) | 15.25 ± 1.35 | 17.27 ± 1.30 | 15.25 ± 1.08 | 34.96 | **< 0.001** |

Values are expressed as mean ± standard deviation (SD). A one-way repeated measures ANOVA was performed to determine statistical differences between respiratory protocols. Statistically significant p-values(***p < 0.001) are presented in bold. HR: Hearth Rate.

revealed statistically significant differences between protocols for final heart rate (p < 0.000) and perceived exertion (p < 0.000), but no significant difference in the time to return to baseline (p = 0.128) (Group-level differences in final heart rate, recovery time, and perceived exertion across protocols are visualized in Fig 4 A-B-C).

## Post-hoc analysis

Post-hoc pairwise comparisons using Tukey's HSD test are summarized in (Tables 4, 5).

Box breathing resulted in a significantly higher final heart rate compared to 6 bpm (p < 0.001).

Box breathing also resulted in significantly higher perceived exertion compared to both 6 bpm (p < 0.001) and spontaneous breathing (p < 0.001).

No significant difference was observed between 6 bpm and spontaneous breathing in terms of final heart rate (p = 0.179) or perceived exertion (p = 1.000).

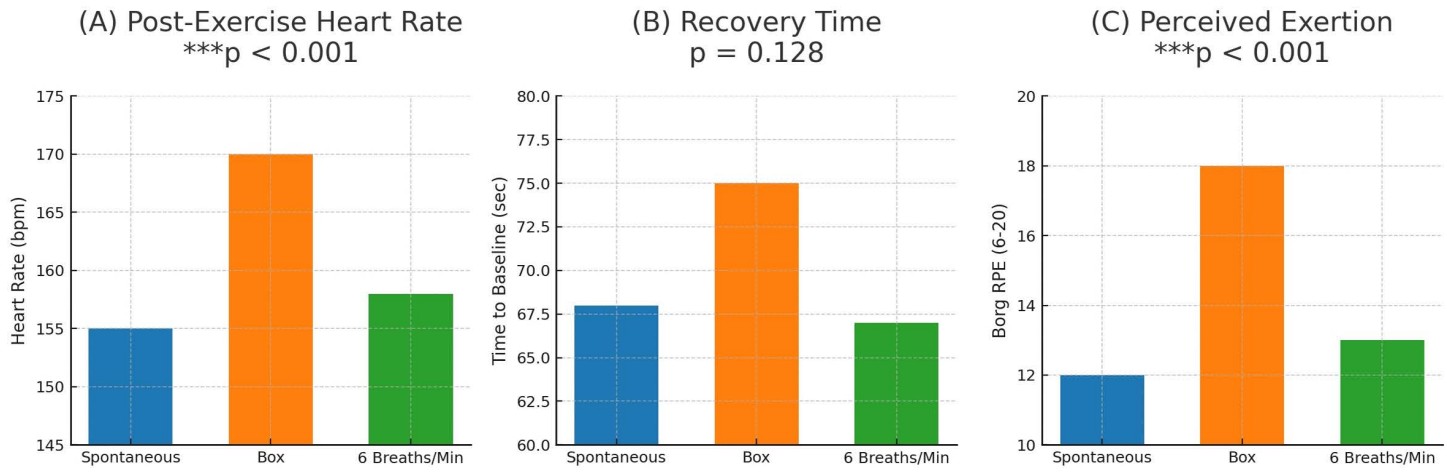

**Fig 4. A-B-C. Heart Rate, Recovery Time and Perceived Effort in Different Breathing Protocols.** Final Heart Rate: The one-way repeated measures ANOVA revealed a significant main effect of breathing protocol on final heart rate (p < 0.001). Post-hoc tests showed that box breathing resulted in a significantly higher final heart rate than 6 bpm (p < 0.001). The difference between box breathing and spontaneous breathing approached, but did not reach, statistical significance (p = 0.054). No significant difference was observed between 6 bpm and spontaneous breathing (p = 0.179). **(B)** Time to Return to Baseline Heart Rate: No statistically significant main effect of breathing protocol was found on recovery time (p = 0.128). However, a non-significant trend toward prolonged recovery was observed following the box breathing condition. **(C)** Perceived Effort (Borg Scale): A significant main effect of breathing protocol was found for perceived exertion (p < 0.001). Post-hoc comparisons indicated that perceived effort was significantly higher after box breathing compared to both the 6 bpm and spontaneous breathing protocols (both p < 0.001). No significant difference was found between 6 bpm and spontaneous breathing (p = 1.000). Note: Data are presented as mean ± SD. Differences were analyzed using a one-way repeated-measures ANOVA with Tukey's HSD post-hoc test (n = 40).

**Table 4. Post-Hoc Test (Tukey HSD) Results.**

| Variable | Comparison | Mean Difference | 95%CI | p-value |
|---|---|---|---|---|
| Final HR (bpm) | Box Breathing vs 6 bpm | +9.87 | [6.32, 13.42] | **< 0.001** |
| | 6 bpm vs Spont. | +4.28 | [-1.97,10.53] | 0.179 |
| | Box Breathing vs Spont. | −5.60 | [-11.32, 0.12] | 0.054 |
| Perceived Exertion (Borg Scale) | Box Breathing vs 6 bpm | −2.03 | [-2.51, -1.55] | **< 0.001** |
| | 6 bpm vs Spontaneous | 0.00 | [-0.57, +0.57] | 1.000 |
| | Box Breathing vs Spont. | +2.03 | [+1.48, +2.58] | **< 0.001** |

Post-hoc pairwise comparisons were performed using Tukey's HSD test to identify specific differences between respiratory protocols. Statistically significant p-values (p < 0.001) are presented in bold. Spont.: Spontaneous.

**Table 5. Effect Sizes: η² (ANOVA) and Cohen's d (Pairwise Comparisons).**

| Variable | η² (95% CI) | Spont. vs Box [95% CI]) | Spont. vs 6 bpm [95% CI] | Box vs 6 bpm [95% CI]) |
|---|---|---|---|---|
| Final HR (bpm) | 0.127 (0.08-0.18) | −0.565 (−0.82 to −0.31) | 0.378 (0.13 to 0.63) | 0.907 (0.64 to 1.17)* |
| Return Time to Baseline (s) | 0.034 (0.00-0.07) | −0.280 (−0.53 to −0.03) | 0.151 (−0.10 to 0.40) | 0.502 (0.25 to 0.76) |
| Borg Scale (RPE) | 0.374 (0.31-0.44)* | −1.525 (−1.82 to −1.23)* | 0.000 (−0.25 to 0.25) | 1.694 (1.41 to 1.98)* |

**Notes:** η² values represent ANOVA effect sizes (small: 0.01, medium: 0.06, large: ≥ 0.14). Cohen's d values show standardized mean differences (small: 0.2, medium: 0.5, large: ≥ 0.8). Asterisks (*) denote effect sizes exceeding conventional large-effect thresholds. All values rounded to 3 decimal places; 95% confidence intervals in parentheses. Spont.: Spontaneous, Box: Box Breathing.

## 4 Discussion

The aim of this study was to compare the effects of three different breathing protocols (spontaneous breathing, box breathing, and 6 bpm) on heart rate, recovery time, and perceived exertion after high-intensity interval training (HIIT) on a spin bike. The findings revealed that the Box Breathing protocol resulted in significantly higher post-exercise heart rates and perceived exertion scores compared to the other protocols. These results indicate that the choice of breathing technique can significantly influence both physiological and perceptual responses during post-exercise recovery.

Contrary to the typical parasympathetic activation associated with slow breathing techniques [1,20], the Box Breathing protocol elicited a significantly higher post-exercise heart rate compared to both 6 bpm and spontaneous breathing. A potential explanation for this paradoxical response lies in the unique structure of box breathing, which incorporates breath-holds. During the high metabolic demand following HIIT, these apneic phases may have induced transient hypercapnia, thereby stimulating a sympathetic response [21] a mechanism supported by findings that breath-holding can temporarily reduce heart rate variability post-exercise [20]. This suggests that the autonomic effects of breathing techniques are context-dependent and can be significantly modulated by exercise intensity. Future research incorporating direct measurements of end-tidal $CO_2$ and heart rate variability is warranted to confirm this hypothesis.

The observed differences were substantiated by large effect sizes for both final heart rate (η² = 0.127; Box vs. 6 bpm Cohen's d = 0.907) and perceived exertion (η² = 0.374; Box vs. Spont. d = 1.525, Box vs. 6 bpm d = 1.694), underscoring the substantial physiological and perceptual impact of the box breathing technique. In contrast, no significant difference was found in recovery time to baseline heart rate between protocols, which contrasts with studies reporting facilitatory effects of similar techniques on recovery [22,23]. This discrepancy may be attributed to differences in exercise intensity, participant characteristics, or protocol duration, highlighting the need for future studies with standardized long-term applications

on homogeneous groups to clarify these effects. The moderate effect size for recovery time between Box Breathing and 6 bpm (Cohen's d = 0.502), though statistically non-significant, warrants further investigation into the potential benefits of rhythmic breathing like 6 bpm.

The 6 bpm protocol, a form of resonance frequency breathing known to enhance cardiorespiratory synchronization [2,14], resulted in heart rate and perceived exertion levels comparable to spontaneous breathing. This aligns with previous reports that slow breathing may not invariably accelerate physiological recovery post-exercise [9], suggesting its primary benefit in this context may not be superior cardiovascular restoration compared to natural breathing.

A particularly striking finding was the substantially higher perceived exertion following Box Breathing, supported by a very large effect size ($\eta^2 = 0.374$). The breath-hold components of this technique, performed under the high physiological stress of post-HIIT recovery, likely contributed to increased psychological and physical strain [11,24]. This contrasts with studies conducted in low-stress or low-intensity conditions [2], underscoring that the perceptual impact of breathing techniques is highly sensitive to the concurrent physiological state.

Finally, there are only a limited number of studies in the literature that directly evaluate the acute effects of breathing techniques on post-exercise physiological recovery [12,15]. Additionally, Garcia et al. [25] highlighted that individuals with lower resting heart rate (RHR) exhibited faster HRR and stronger parasympathetic reactivation, while Danieli et al. [26] emphasized that resting HRV values in middle-aged individuals were significantly positively correlated with early HRR post-exercise [27]. These findings draw attention to the effects of resting heart rate and age on autonomic regulation and suggest the need for age-specific analyses.

In this context, our study contributes significantly to the field by comparing the effects of controlled breathing techniques on post-exercise heart rate and perceived exertion in young, physically active individuals.

## 5. Conclusions and recommendations

The results of this study strengthen the assessment of different breathing protocols as indicators that may create significant differences in physiological recovery and perceived exertion after high-intensity interval training (HIIT).

**Compared to spontaneous breathing**

- The Box Breathing protocol resulted in higher heart rate and perceived exertion.

- The 6 bpm protocol led to lower physiological and perceptual loads.

- Although there was no significant difference in recovery time, there was a trend toward delayed recovery under the Box Breathing condition.

These results suggest that controlled breathing techniques may not produce uniform physiological responses in all individuals and that their effects may vary depending on the type and intensity of exercise.

**Limitations and future studies**

This study has several limitations that should be considered. Firstly, the findings are constrained by their acute nature, capturing only the immediate effects of a single application of the breathing protocols. The long-term adaptations and efficacy of sustained practice (e.g., over 8–12 weeks) remain unknown and represent a crucial area for future investigation.

Secondly, the generalizability of the results is limited by the specific sample characteristics. The participants were young, healthy, and physically active sports science students. Consequently, these findings may not be directly applicable to older adults, elite athletes, sedentary individuals, or populations with different fitness levels or health conditions. The effectiveness of these breathing strategies is likely influenced by factors such as age, baseline fitness, and training status, which warrants further exploration.

To address these limitations, future chronic studies should monitor changes in parameters such as HR recovery and heart rate variability (HRV), incorporating additional physiological markers like baroreflex sensitivity [3]. Furthermore, research should compare the effects of these protocols across different exercise intensities, durations, and population groups. The use of respiratory rate monitors to standardize breathing patterns would also enhance the reliability of future data.

### Practical implications

The findings of this study offer valuable practical applications for coaches, athletes, and fitness professionals. According to our results, the 6 breaths/min breathing protocol (5-second inhale, 5-second exhale) is a more effective and less stressful strategy for post-HIIT recovery compared to Box Breathing (4-4-4-4). This technique can be easily implemented in training settings: coaches can instruct athletes to begin the 6 breaths/min rhythm during rest intervals between sets or immediately upon finishing a session, using simple verbal cues ("Breathe in... two... three... four... five... Now breathe out... two... three... four... five...") or a visual metronome to maintain the pace. The protocol's simplicity allows for rapid adoption without the need for specialized equipment. In contrast, practitioners should be cautious when prescribing Box Breathing immediately after high-intensity exercise, as it may inadvertently increase physiological and perceptual stress, potentially delaying recovery. For optimal results, the regular application of the 6 breaths/min technique during inter-set rest periods or post-training is recommended.

### Supporting information

**S1 Data. All Data.**
(XLSX)

**S1 File. S1 Statistical Summary.**
(XLSX)

### Author contributions

**Conceptualization:** Murat Kasap.

**Data curation:** Murat Kasap.

**Formal analysis:** Murat Kasap, Gökhan Recep AYDIN.

**Methodology:** Murat Kasap.

**Visualization:** Gökhan Recep AYDIN.

**Writing – original draft:** Murat Kasap.

**Writing – review & editing:** Gökhan Recep AYDIN.

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
