## [Decision Letter · Decision Letter 0]

25 Sep 2025

PONE-D-25-41258Box breathing or six breaths per minute: which strategy improves athletes post-HIIT cardiovascular recovery?PLOS ONE?

Dear Dr. Kasap,

Thank you for submitting your manuscript to PLOS ONE. After careful consideration, we feel that it has merit but does not fully meet PLOS ONE’s publication criteria as it currently stands. Therefore, we invite you to submit a revised version of the manuscript that addresses the points raised during the review process.

We look forward to receiving your revised manuscript.

Kind regards,

Ozeas de Lima Lins-Filho, Ph.D.

Academic Editor

PLOS ONE

Journal Requirements:

4. Please remove your figures from within your manuscript file, leaving only the individual TIFF/EPS image files, uploaded separately. These will be automatically included in the reviewers’ PDF.

Additional Editor Comments: 

Dear Dr. Murat Kasap

Your manuscript entitled "Box breathing or six breaths per minute: which strategy improves athletes post-HIIT cardiovascular recovery? submitted to PLOS One has been reviewed. The comments of the reviewers are included at the bottom of this letter.

The Reviewers found your manuscript interesting but it will require Major revision in order to be considered for publication. Therefore, I invite you to respond to the Reviewers' comments and revise your manuscript.

Reviewer #1:

The study sought to investigate a relevant topic by comparing the impact of three different breathing protocols on cardiovascular and perceptual recovery. The proposal is interesting, as it addresses an area of growing scientific and practical interest, especially in the context of physical performance and health. However, some adjustments are necessary to improve the clarity and consistency of the text.

General Recommendations

References should be formatted according to the Vancouver style, as required by the journal.

Objectives and hypotheses should be presented within the running text, rather than in separate subsections.

Introduction

Line 43: The sentence “...also how and how quickly the body responds to this load..” contains a repeated “how” and should be corrected.

Line 52: The term “psychological variables” requires clarification. Please specify which variables are considered (e.g., stress) and provide a brief description.

Materials and Methods

Exclusion criteria were not reported. If applied, please describe them.

Was there any control regarding the type of sport practiced by participants? Certain sports involving respiratory techniques could bias the sample.

Please provide more details about the randomization process, including the method used.

Clarify how respiratory control was adequately ensured during the procedures.

Were heart rate variability (HRV) data collected? If so, their inclusion could substantially enrich the discussion.

Results

Measures of dispersion for the mean difference are missing in Table 4.

Discussion

The paragraph addressing heart rate variability states:

“In addition, the difference detected in heart rate variability was found to be significant not only statistically but also in terms of effect size (η² = 0.127). In particular, Cohen's d = 0.907 obtained in the comparison between Box Breathing and 6 Breaths/min indicates a large difference. This supports the idea that the aforementioned breathing technique may have a significant effect on physiological stress.”

This section is confusing.

It is necessary to clarify whether HRV data were actually collected or whether only heart rate (HR) was measured. HRV indices, such as time-domain and frequency-domain parameters, could strengthen the interpretation of the findings.

If only HR was measured, please revise the terminology to avoid misleading readers by conflating HR with HRV.

Reviewer #2:

This manuscript presents an experimental study comparing the effects of two controlled breathing strategies—box breathing and breathing at six cycles per minute (6 bpm)—on cardiovascular and perceptual recovery after HIIT sessions on a cycle ergometer. This is a current topic of practical relevance for sports training and exercise physiology, especially considering the scarcity of studies on respiratory protocols applied to acute recovery after high-intensity exercise.

The study is well-structured, with an appropriate methodological design (randomized crossover), an adequate sample size (n = 40), and statistical analysis consistent with the proposed objectives (repeated measures ANOVA, Tukey's post-hoc test, sample power analysis, and effect size calculation). The article also meets the formal criteria required by PLOS ONE, such as sections on ethics, conflicts of interest, and data availability.

However, for the manuscript to fully meet the publication criteria, adjustments are needed regarding methodological clarity, standardization of language, and a more in-depth discussion of practical aspects and limitations.

Strengths

The topic is relevant and applied, with direct implications for optimizing recovery between HIIT sessions, a critical factor for athletic performance and safety in training prescription.

The randomized crossover experimental design is adequate to minimize interindividual variability, increasing the reliability of the findings.

The sample size (n = 40) is satisfactory, and the inclusion of power analysis via G*Power demonstrates methodological rigor.

The presentation of the results through tables and figures is well-designed, facilitating comprehension of the data and effectively complementing the text.

Points to Address

1. Determination of HRmax

The manuscript states that participants performed HIIT based on 85–95% of their maximum heart rate (HRmax), but does not specify how this HRmax was determined. It is essential to clarify whether a direct exercise stress test or a predictive formula (e.g., 220 - age) was used, as this directly impacts the accuracy of the exercise prescription.

2. Standardization of Terminology

There are inconsistencies in how the six-cycles-per-minute breathing protocol is referred to: "six breaths per minute," "6 breaths/min," "6 breaths/minute." It is suggested to standardize this to "6 bpm" (breaths per minute) throughout the manuscript, maintaining consistency and clarity.

3. Discussion – Practical Applicability

The discussion could be improved with greater emphasis on the practical implications of the findings, especially for coaches, physical trainers, and physiologists. For example, how can the 6 bpm breathing protocol be easily implemented in sports settings? What is the best way to instruct athletes? Should it be applied immediately after exercise or in separate sessions?

4. Limitations

The study acknowledges limitations such as its focus on acute effects and the sample size of young university students. However, it is recommended to emphasize that the generalizability of the results is limited and that the effects may vary depending on the intensity of the exercise, the sport, and the training level of the participants.

5. Redundancy and Style

The discussion presents some repetitions and could be more concise in certain sections. A revision is suggested for greater objectivity, without compromising the depth of the analysis.

Summary Recommendations

Include clear details on the method for determining HRmax.

Standardize the nomenclature to "6 bpm" throughout the text.

Streamline the discussion, reducing repetitions.

Expand the discussion on the practical applicability of respiratory protocols.

Reinforce the limitations related to the sample and the acute nature of the observed effects.

Final Comment

This manuscript presents a relevant and well-conducted contribution to the field of exercise physiology, exploring understudied respiratory strategies in the context of post-HIIT recovery. The suggested adjustments are specific and aim to improve methodological clarity, textual standardization, and the applied value of the discussion. With these modifications, the study has the potential for publication in PLOS ONE.

Reviewers' comments:

Reviewer's Responses to Questions

**Comments to the Author**

1. Is the manuscript technically sound, and do the data support the conclusions?

Reviewer #1: Yes

Reviewer #2: Yes

2. Has the statistical analysis been performed appropriately and rigorously?

Reviewer #1: Yes

Reviewer #2: Yes

3. Have the authors made all data underlying the findings in their manuscript fully available?

Reviewer #1: Yes

Reviewer #2: Yes

4. Is the manuscript presented in an intelligible fashion and written in standard English?

Reviewer #1: Yes

Reviewer #2: Yes

Reviewer #1: The study sought to investigate a relevant topic by comparing the impact of three different breathing protocols on cardiovascular and perceptual recovery. The proposal is interesting, as it addresses an area of growing scientific and practical interest, especially in the context of physical performance and health. However, some adjustments are necessary to improve the clarity and consistency of the text.

General Recommendations

References should be formatted according to the Vancouver style, as required by the journal.

Objectives and hypotheses should be presented within the running text, rather than in separate subsections.

Introduction

Line 43: The sentence “...also how and how quickly the body responds to this load..” contains a repeated “how” and should be corrected.

Line 52: The term “psychological variables” requires clarification. Please specify which variables are considered (e.g., stress) and provide a brief description.

Materials and Methods

Exclusion criteria were not reported. If applied, please describe them.

Was there any control regarding the type of sport practiced by participants? Certain sports involving respiratory techniques could bias the sample.

Please provide more details about the randomization process, including the method used.

Clarify how respiratory control was adequately ensured during the procedures.

Were heart rate variability (HRV) data collected? If so, their inclusion could substantially enrich the discussion.

Results

Measures of dispersion for the mean difference are missing in Table 4.

Discussion

The paragraph addressing heart rate variability states:

“In addition, the difference detected in heart rate variability was found to be significant not only statistically but also in terms of effect size (η² = 0.127). In particular, Cohen's d = 0.907 obtained in the comparison between Box Breathing and 6 Breaths/min indicates a large difference. This supports the idea that the aforementioned breathing technique may have a significant effect on physiological stress.”

This section is confusing.

It is necessary to clarify whether HRV data were actually collected or whether only heart rate (HR) was measured. HRV indices, such as time-domain and frequency-domain parameters, could strengthen the interpretation of the findings.

If only HR was measured, please revise the terminology to avoid misleading readers by conflating HR with HRV.

Reviewer #2: Overview

This manuscript presents an experimental study comparing the effects of two controlled breathing strategies—box breathing and breathing at six cycles per minute (6 bpm)—on cardiovascular and perceptual recovery after HIIT sessions on a cycle ergometer. This is a current topic of practical relevance for sports training and exercise physiology, especially considering the scarcity of studies on respiratory protocols applied to acute recovery after high-intensity exercise.

The study is well-structured, with an appropriate methodological design (randomized crossover), an adequate sample size (n = 40), and statistical analysis consistent with the proposed objectives (repeated measures ANOVA, Tukey's post-hoc test, sample power analysis, and effect size calculation). The article also meets the formal criteria required by PLOS ONE, such as sections on ethics, conflicts of interest, and data availability.

However, for the manuscript to fully meet the publication criteria, adjustments are needed regarding methodological clarity, standardization of language, and a more in-depth discussion of practical aspects and limitations.

Strengths

The topic is relevant and applied, with direct implications for optimizing recovery between HIIT sessions, a critical factor for athletic performance and safety in training prescription.

The randomized crossover experimental design is adequate to minimize interindividual variability, increasing the reliability of the findings.

The sample size (n = 40) is satisfactory, and the inclusion of power analysis via G*Power demonstrates methodological rigor.

The presentation of the results through tables and figures is well-designed, facilitating comprehension of the data and effectively complementing the text.

Points to Address

1. Determination of HRmax

The manuscript states that participants performed HIIT based on 85–95% of their maximum heart rate (HRmax), but does not specify how this HRmax was determined. It is essential to clarify whether a direct exercise stress test or a predictive formula (e.g., 220 - age) was used, as this directly impacts the accuracy of the exercise prescription.

2. Standardization of Terminology

There are inconsistencies in how the six-cycles-per-minute breathing protocol is referred to: "six breaths per minute," "6 breaths/min," "6 breaths/minute." It is suggested to standardize this to "6 bpm" (breaths per minute) throughout the manuscript, maintaining consistency and clarity.

3. Discussion – Practical Applicability

The discussion could be improved with greater emphasis on the practical implications of the findings, especially for coaches, physical trainers, and physiologists. For example, how can the 6 bpm breathing protocol be easily implemented in sports settings? What is the best way to instruct athletes? Should it be applied immediately after exercise or in separate sessions?

4. Limitations

The study acknowledges limitations such as its focus on acute effects and the sample size of young university students. However, it is recommended to emphasize that the generalizability of the results is limited and that the effects may vary depending on the intensity of the exercise, the sport, and the training level of the participants.

5. Redundancy and Style

The discussion presents some repetitions and could be more concise in certain sections. A revision is suggested for greater objectivity, without compromising the depth of the analysis.

Summary Recommendations

Include clear details on the method for determining HRmax.

Standardize the nomenclature to "6 bpm" throughout the text.

Streamline the discussion, reducing repetitions.

Expand the discussion on the practical applicability of respiratory protocols.

Reinforce the limitations related to the sample and the acute nature of the observed effects.

Final Comment

This manuscript presents a relevant and well-conducted contribution to the field of exercise physiology, exploring understudied respiratory strategies in the context of post-HIIT recovery. The suggested adjustments are specific and aim to improve methodological clarity, textual standardization, and the applied value of the discussion. With these modifications, the study has the potential for publication in PLOS ONE.

**Do you want your identity to be public for this peer review?** For information about this choice, including consent withdrawal, please see our Privacy Policy

Reviewer #1: **Yes: ** José Lucas Porto Aguair

Reviewer #2: No

---

## [Author Response · Author response to Decision Letter 1]

14 Oct 2025

Response to Reviewers

We would like to thank the Academic Editor and both Reviewers for their valuable comments and suggestions. These comments significantly improved the quality and clarity of our article.

Below, we provide point-by-point responses to each comment.

Reviewer 1's comments are highlighted in yellow, and Reviewer 2's comments are highlighted in turquoise, followed by our responses and a summary of the changes made (Previous version, Edited version).

We hope that the revised manuscript meets the expectations of the reviewers and the journal. We sincerely appreciate the opportunity to revise and resubmit our work.

Kind regards,

Dr. Murat Kasap

Corresponding Author

Reviewer #1:

1) References should be formatted according to the Vancouver style, as required by the journal.

Previous version

1a) “The post-exercise recovery process plays a critical role in increasing training efficiency, maintaining high-level performance, and reducing the risk of injury in athletes (Zaccaro et al., 2018; Kumar et al., 2024)…….”

References List

1b)

Zaccaro, A., Piarulli, A., Laurino, M., Garbella, E., Menicucci, D., Neri, B., & Gemignani, A. (2018). How breath-control can change your life: A systematic review on psycho-physiological correlates of slow breathing. Front. Hum. Neurosci., 12, 353. https://doi.org/10.3389/fnhum.2018.00353

1c)

Kumar, S., Das, R., & Sharma, V. (2024). Effects of 6 breaths-per-minute breathing and box breathing techniques on anxiety, perceived stress, and physiological indicators: A randomized controlled trial. Complement. Ther. Clin. Pract., 54, 102150. https://doi.org/10.1016/j.ctcp.2024.102150

Edited version

1a) The post-exercise recovery process plays a critical role in increasing training efficiency, maintaining high-level performance, and reducing the risk of injury in athletes [1, 2].

References List

1b)

1. Zaccaro A, Piarulli A, Laurino M, Garbella E, Menicucci D, Neri B, et al. How breath-control can change your life: A systematic review on psycho-physiological correlates of slow breathing. Front Hum Neurosci. 2018;12:353. https://doi.org/10.3389/fnhum.2018.00353

1c)

2. Kumar S, Das R, Sharma V. Effects of 6 breaths-per-minute breathing and box breathing techniques on anxiety, perceived stress, and physiological indicators: A randomized controlled trial. Complement Ther Clin Pract. 2024;54:102150. https://doi.org/10.1016/j.ctcp.2024.102150

Note: The "Vancouver style" correction given above has been applied to the entire article.

2) Objectives and hypotheses should be presented within the running text, rather than in separate subsections.

Previous version

2a)…….. perceived exertion following high-intensity interval training (HIIT) are quite limited.

Study objective and hypotheses

Building on this gap, this study aimed to investigate the acute effects of three different breathing protocols (Box Breathing, 6 Breaths per Minute, Spontaneous Breathing) on physiological recovery during high-intensity interval (HIIT) cycling exercise on a spin bike in sports science students of similar age and physical capacity.

Hypotheses:

H1: Box Breathing and 6 Breaths per Minute protocols will positively affect heart rate recovery and perceived exertion (Borg RPE scores) compared to spontaneous breathing.

H2: Controlled breathing techniques will reduce perceived exertion (Borg RPE scores) more effectively than spontaneous breathing.

H3: 6 Breaths per Minute may have a stronger effect on cardiovascular and perceptual recovery metrics than the Box Breathing protocol.

Edited version

The hypotheses tested in the study were as follows: H1: Box Breathing and 6 Breaths per Minute protocols will positively affect heart rate recovery and perceived exertion (Borg RPE scores) compared to spontaneous breathing. H2: Controlled breathing techniques will reduce perceived exertion (Borg RPE scores) more effectively than spontaneous breathing. H3: 6 Breaths per Minute may have a stronger effect on cardiovascular and perceptual recovery metrics than the Box Breathing protocol.

Note: Objectives and hypotheses are presented in both the summary and introduction sections as a continuation of the text.

3) Introduction

3a) Line 43: The sentence “...also how and how quickly the body responds to this load..” contains a repeated “how” and should be corrected.

Previous version

3a) Modern training science considers recovery to be one of the fundamental determinants of performance, taking into account not only the training load but also how and how quickly the body responds to this load.

Edited version

3a) Modern training science considers recovery to be one of the fundamental determinants of performance, taking into account not only the training load but also the manner and speed with which the body responds to it."

3b) Line 52: The term “psychological variables” requires clarification. Please specify which variables are considered (e.g., stress) and provide a brief description.

Previous version

Studies on the physiological effects of breathing exercises have reported positive results in cardiopulmonary parameters such as increased vagal tone, improved heart rate variability (HRV), and improved baroreflex sensitivity (Lehrer & Gevirtz, 2014; Laborde, 2019). Slow and controlled breathing suppresses sympathetic activity and increases parasympathetic responses, leading to significant improvements in heart rate, blood pressure, and psychological stress levels (Morgan et al., 2024; Hopper et al., 2019).

Edited version

"Studies on the physiological effects of breathing exercises have reported positive results in cardiopulmonary parameters such as increased vagal tone, improved heart rate variability (HRV), and improved baroreflex sensitivity [7,8]. Slow and controlled breathing suppresses sympathetic activity and increases parasympathetic responses, leading to significant improvements in heart rate, and blood pressure. This physiological shift is directly associated with a reduction in psychological stress, which in this context refers to measurable decreases in perceived stress and state anxiety, as well as an improvement in overall emotional well-being [9,10,14]."

4) Materials and Methods

a) Exclusion criteria were not reported. If applied, please describe them.

b) Exclusion criteria were not reported. If applied, please describe them.

c) Was there any control regarding the type of sport practiced by participants? Certain sports involving respiratory techniques could bias the sample.

d) Please provide more details about the randomization process, including the method used.

e) Clarify how respiratory control was adequately ensured during the procedures.

f) Were heart rate variability (HRV) data collected? If so, their inclusion could substantially enrich the discussion.

Previous version

Participants

Forty students (25 males, 15 females; age: 20.95 ± 1.75 years; BMI: 22.97 ± 3.25 kg/m²) from the Faculty of Sports Sciences at Bartın University voluntarily participated in the study. Participants were individuals who engaged in at least 2 days per week of physical activity at a level that included aerobic or anaerobic exercise. All participants were familiar with bicycle ergometer training and had no known cardiopulmonary or orthopaedic conditions. The study was approved by the Bartın University Ethics Committee (Approval No: 2025-SBB-0244) and conducted in accordance with the Helsinki Declaration. All participants were of legal age and signed written informed consent before the study.

Application procedure

A randomized crossover design was employed in the study. Each participant……

Edited version

Participants

4 a-b-c) Forty students from Bartın University's Faculty of Sports Sciences (25 male, 15 female; age: 20.95 ± 1.75 years; BMI: 22.97 ± 3.25 kg/m²) voluntarily participated in the study. The participants were athletes who engaged in physical activity comprising at least two days of aerobic or anaerobic exercise per week, had no prior experience with breathing exercises or meditation-based activities, and were not currently practicing them. All participants were familiar with cycle ergometer training and had no known cardiopulmonary or orthopedic disorders. The participants were not practicing any techniques such as single-breath diving, breathing exercises, or meditation. The study was approved by the Bartın University Ethics Committee (Approval No: 2025-SBB-0244) and conducted in accordance with the Declaration of Helsinki. All participants were of legal age and provided written informed consent prior to the study.

Inclusion Criteria:

-Voluntary participation.

-Age 18-24.

-Student in the Faculty of Sports Sciences.

-Must not have previously engaged in, and must not be currently engaging in, any breathing exercises or meditation-based activities.

Exclusion Criteria:

-Failure to meet one or more of the inclusion criteria listed above was defined as an exclusion criterion.

4 d) Randomization

The order of the three breathing protocols was randomized for each participant using a computer-generated random sequence "(www.randomizer.org)."

The order was as follows:

Session 1

37, 36, 11, 27, 15, 38, 1, 29, 24, 7, 4, 2, 34, 23, 33, 6, 14, 5, 40, 21, 12, 32, 13, 3, 16, 25, 31, 35, 18, 26, 30, 17, 20, 9, 8, 28, 19, 39, 10, 22

Session 2

32, 2, 19, 37, 33, 11, 15, 21, 29, 39, 28, 6, 40, 31, 18, 27, 22, 16, 3, 38, 12, 13, 26, 4, 30, 35, 36, 17, 34, 5, 20, 10, 24, 7, 25, 14, 23, 8, 9, 1

Session 3

5, 12, 15, 10, 28, 13, 4, 36, 25, 31, 18, 19, 2, 7, 35, 39, 32, 3, 40, 14, 26, 23, 22, 33, 9, 38, 16, 29, 8, 17, 6, 34, 11, 20, 1, 37, 24, 30, 21, 27

4 e) Clarify how respiratory control was adequately ensured during the procedures.

Specially designed video slides for the three distinct breathing protocols were projected onto the wall in front of the participants in the spin bike room. The test flow guided the athletes both visually and auditorily.

Figure 1. Example (4-4-4-4) square breathing video presentation stage visuals

Respiratory control was rigorously ensured through a standardized, audiovisual guidance system that minimized participant discretion and ensured strict adherence to the prescribed breathing protocols. The procedure was as follows:

Specially designed video slides for each of the three distinct breathing protocols were projected onto the wall in front of the participants. The entire test flow was governed by this system, which provided simultaneous visual and auditory commands.

Prior to each testing session, participants were familiarized with the specific breathing protocol they would perform to ensure comprehension.

The control process for a single cycle (e.g., the 4-4-4-4 square breathing protocol) was as follows:

1- The session began with a 30-second exercise bout initiated by an audible "start test" command. A countdown timer was displayed. At the end of the exercise, an auditory cue announced the transition to a 60-second rest/breathing period. During this period, each phase of the breathing cycle was precisely guided:

2-3-4-5 An audible "inhale" cue was given, accompanied by a visual timer counting down 4 seconds. This was sequentially followed by "hold breath" (4 seconds), "exhale" (4 seconds), and again "hold breath" (4 seconds). This cycle was repeated throughout the 60-second rest period. A verbal and visual warning was given at the 55-second mark to prepare for the next exercise bout.

Moreover "Researchers visually monitored each participant's chest movement and provided immediate verbal feedback to ensure adherence to the target breathing rhythm throughout the recovery period."

The sequence of 30-second exercise followed by the 60-second guided breathing period was repeated until the total session duration of 15 minutes was completed.

6-The test was terminated automatically upon reaching the 15-minute mark with an audible "test finished" command, immediately followed by the display of the Borg Scale for participant rating.

This method of automated, step-by-step audiovisual guidance ensured that all participants followed the exact same rhythm and timing for each breath, thereby guaranteeing adequate and consistent respiratory control throughout the procedures.

4f) Were heart rate variability (HRV) data collected? If so, their inclusion could substantially enrich the discussion.

Previous version

This mechanism differs from findings in low-intensity studies and suggests that the effects of breathing techniques may vary depending on the exercise modality. Future studies should validate this hypothesis using end-tidal CO₂ and HRV measurements.

In addition, the difference detected in heart rate variability was found to be significant not only statistically but also in terms of effect size (η² = 0.127). In particular, Cohen's d = 0.907 obtained in the comparison between Box Breathing and 6 Breaths/min indicates a large difference. This supports the idea that the aforementioned breathing technique may have a significant effect on physiological stress.

Edited version

"Were heart rate variability (HRV) data collected?" The paragraph was updated to avoid ambiguity, and the Parameters Measured and Observed in the Study section was added to the Materials and methods section.

The observed differences were substantiated by large effect sizes for both final heart rate (η² = 0.127; Box vs. 6 bpm Cohen's d = 0.907) and perceived exertion (η² = 0.374; Box vs. Spont. d = 1.525, Box vs. 6 bpm d = 1.694), underscoring the substantial physiological and perceptual impact of the box breathing technique. In contrast, no significant difference was found in recovery time to baseline heart rate between protocols, which contrasts with studies reporting facilitatory effects of similar techniques on recovery [21,23]. This discrepancy may be attributed to differences in exercise intensity, participant characteristics, or protocol duration, highlighting the need for future studies with standardized long-term applications on homogeneous groups to clarify these effects. The moderate effect size for recovery time between Box Breathing and 6 bpm (Cohen's d = 0.502), though statistically non-significant, warrants further investigation into the potential benefits of rhythmic breathing like 6 bpm.

Measured and Observed Parameters

The following primary outcome variables were assessed for each breathing protocol:

• Initial Heart Rate (HR): HR measured at the end of the 10-minute warm-up period.

• Post-Exercise Heart Rate (HR): HR measured immediately upon completion of the final HIIT interval.

• Heart Rate Recovery Time: The time elapsed from the cessation of exercise until HR returned to the individual's pre-exercise baseline (Initial HR) value.

• Rating of Perceived Exertion (RPE): Assessed using the Borg RPE scale (6-20) immediately after each test session.

5 Result

Measures of dispersion for the mean difference are missing in Table 4.

Previous version

Table 4. Post-Hoc Test (Tukey HSD) Results

Variable Comparison Mean Difference p-value

Final Heart Rate (beats/min) 6 Breaths vs Box Breathing +9.87 p < 0.001***

6 Breaths vs Control +4.28 0.179

Box Breathing vs Control -5.60 0.054

Perceived Exertion (Borg Scale) 6 Breaths vs Box Breathing -2.03 p < 0.001***

6 Breaths vs Control 0.00 1.000

Box Breathing vs Control +2.03 p < 0.001***

Edited version

Table 4. Post-Hoc Test (Tukey HSD) Results

Variable Comparison Mean Difference 95%CI p-value

Final HR (bpm) Box Breathing vs 6 bpm +9.87 [6.32, 13.42] < 0.001

6 bpm vs Spont. +4.28 [-1.97,10.53] 0.179

Box Breathing vs Spont. -5.60 [-11.32, 0.12] 0.054

Perceived Exertion (Borg Scale) Box Breathing vs 6 bpm -2.03 [-2.51, -1.55] < 0.001

6 bpm vs Spontaneous 0.00 [-0.57, +0.57] 1.000

Box Breathing vs Spont. +2.03 [+1.48, +2.58] < 0.001

Post-hoc pairwise comparisons were performed using Tukey's HSD test to identify specific differences between respiratory protocols. Statistically significant p-values (p < 0.001) are presented in bold. Spont.: Spontaneous.

6 Discussion

The paragraph addressing heart rate variability states:

“In addition, the difference detected in heart rate variability was found to be significant not only statistically but also in terms of effect size (η² = 0.127). In particular, Cohen's d = 0.907 obtained in the com

---

## [Editor Report · Decision Letter 1]

29 Oct 2025

Box breathing or six breaths per minute: which strategy improves athletes post-HIIT cardiovascular recovery?

PONE-D-25-41258R1

Dear Dr. Murat Kasap,

We’re pleased to inform you that your manuscript has been judged scientifically suitable for publication and will be formally accepted for publication once it meets all outstanding technical requirements.

Kind regards,

Ozeas de Lima Lins-Filho, Ph.D.

Academic Editor

PLOS ONE

Additional Editor Comments (optional):

Reviewers' comments:

No additional comments.

---

## [Editor Report · Acceptance letter]

PONE-D-25-41258R1

PLOS ONE

Dear Dr. Kasap,

I'm pleased to inform you that your manuscript has been deemed suitable for publication in PLOS ONE. Congratulations! Your manuscript is now being handed over to our production team.

Kind regards,

on behalf of

Dr. Ozeas de Lima Lins-Filho

Academic Editor

PLOS ONE